# Elevated Urinary Hepcidin Level and Hypoferremia in Infants with Febrile Urinary Tract Infection: A Prospective Cohort Study

**DOI:** 10.3390/children10050870

**Published:** 2023-05-12

**Authors:** Yu-Chen Hsu, Hsin-Chun Huang, Kuo-Su Tang, Li-Ting Su, Ying-Hsien Huang, Hui-Chen Huang, I-Lun Chen

**Affiliations:** 1Department of Pediatrics, College of Medicine, Kaohsiung Chang Gung Memorial Hospital, Chang Gung University, Kaohsiung 83301, Taiwan; yuchen16k@cgmh.org.tw (Y.-C.H.); drhhuang@hotmail.com (H.-C.H.); tang1004@cgmh.org.tw (K.-S.T.); yhhuang@cgmh.org.tw (Y.-H.H.); flower04_cg@yahoo.com.tw (H.-C.H.); 2School of Medicine, College of Medicine, Chang Gung University, Linkou 33302, Taiwan; 3Antai Medical Care Corporation, Antai Tian-Sheng Memorial Hospital, Pingtung 92842, Taiwan; suliting888@gmail.com; 4School of Traditional Chinese Medicine, College of Medicine, Chang Gung University, Linkou 33302, Taiwan

**Keywords:** hemoglobin, hepcidin, urine, iron, urinary tract infection

## Abstract

To evaluate the kinetics of serum and urinary hepcidin levels along with anemia-related parameters during the infection course of infants with febrile urinary tract infection (UTI), we enrolled febrile infants aged one to four months in this prospective study. Febrile patients with UTI were allocated into *Escherichia coli* (*E. coli*) or non-*E. coli* groups according to urine culture results. Septic workup, blood hepcidin, iron profile, urinalysis, and urinary hepcidin–creatinine ratio were collected upon admission and 3 days after antibiotic treatment. In total, 118 infants were included. On admission, the febrile UTI group showed a significant reduction in serum iron level and a significant elevation of urinary hepcidin–creatinine ratio compared to the febrile control counterpart. Moreover, urinary hepcidin–creatinine ratio had the highest odds ratio, 2.01, in logistics regression analysis. After 3 days of antibiotic treatment, hemoglobin and the urinary hepcidin–creatinine ratio were significantly decreased. Patients with an *E. coli* UTI had a significantly decreased urinary hepcidin–creatinine ratio after 3 days of antibiotics treatment, whereas the non-*E. coli* group showed insignificant changes. Our study suggested that the urinary hepcidin–creatinine ratio elevated during acute febrile urinary tract infection and significantly decreased after 3 days of antibiotics treatment, especially in *E. coli* UTI.

## 1. Introduction

Hepcidin, an antimicrobial peptide first discovered and isolated from urine [1], is produced majorly in the liver and plays a key role in the homeostasis of iron. Hepcidin inhibits iron release from senescent erythrocytes in macrophages, blocks duodenal iron absorption, and causes iron sequestration in the reticuloendothelial system [2]. During infection, hepcidin expression is upregulated by cytokines such as interleukin-6 (IL-6), leading to the characteristic hypoferremia and hyperferritinemia seen in anemia of inflammation. Although hepcidin is a principal regulator in the development of anemia associated with inflammation and infection, there are only a few clinical studies that investigate the kinetics of hepcidin and iron measurements in patients with bacterial infection [3].

Urinary tract infection (UTI) is one of the most common bacterial infectious diseases in young infants. The incidence of UTI in infants <3 months of age for girls is 7.5%, and for uncircumcised boys, 20.7% [4]. Diagnosis of urinary tract infection requires positive urinalysis and urine culture results, in combination with associated history and physical exam findings [5,6]. However, signs and symptoms of UTI in infants are nonspecific, such as fever, vomiting, lethargy, and irritability. Urine culture as the essential evidence for diagnosis of UTI, requires 2–3 days for detection of the responsible microorganism, and is not available in emergencies. In addition, prior studies have demonstrated that the absence of urine white blood cells (WBC) or leucocyte esterase is a common occurrence, observed in at least 10% of pediatric UTI cases [7]. This is particularly evident when the causative organism is not *E. coli* [8]. The conversion of nitrate to nitrite typically requires a four-hour incubation period, rendering nitrite an unreliable marker of UTI in infants, who frequently empty their bladders. Additionally, it is worth noting that not all urinary pathogens possess the ability to reduce nitrate to nitrite, further limiting the usefulness of this diagnostic tool [9].

The urinary tract provides host innate immune defenses through the restriction of iron availability [10,11] and antimicrobial peptides such as defensin, cathelicidin, Tamm-Horsfall protein, lactoferrin, and lipocalin [12]. Although hepcidin also exhibits antimicrobial activity as well and is freely filtered by the glomerulus, few clinical studies have focused on its role in the local innate defense mechanism in the urinary tract system. Recent studies have shed light on the clinical relevance of urinary hepcidin mitigating acute kidney injury by promoting iron sequestration and thereby limiting oxidative stress, free radical damage, and renal injury in cardiac surgery patients [13,14].

UTI is primarily caused by ascending infection rather than hematogenous spread in infants, with *Escherichia coli* (*E. coli*) accounting for approximately 80 percent of cases [15]. In animal studies, urinary hepcidin has been identified as an effective renal antibacterial defense against uropathogenic *E. coli* through the regulation of iron availability, triggering of urinary acidic pH, and direct bacteriostatic action to counter *E. coli* infection [16]. However, clinical studies on the role of urinary hepcidin in infants with *E. coli* UTI are scarce. Hence, the present study aimed to investigate the kinetic changes of serum and urinary hepcidin levels and their relationship with other anemia-associated parameters in infants with febrile UTI. Moreover, we sought to identify potential clinical clues concerning the role of urinary hepcidin in infants with *E. coli* UTI.

## 2. Materials and Methods

### 2.1. Patient Population

This prospective cohort study recruited febrile infants aged 1 to 4 months who were admitted through the pediatric emergency department of a level III hospital between December 2017 and February 2021. We specifically selected infants in this age group to ensure a relatively homogenous population, taking into consideration the known age-related difference in hepcidin [17,18]. In addition, we excluded neonates with UTI as they are often associated with congenital anomalies of the kidney and urinary tract (CAKUT) [19,20]. Furthermore, neonates were not included as changes in Hb levels in this group may be influenced by factors such as polycythemia, hemodilution with somatic growth, alloimmune hemolytic disease of the newborn, or nutritional anemia. The number of infants included was determined by G power software with effect size of 0.6 and power of 0.8. Infants born prematurely, receiving iron supplements, undergoing blood transfusion therapy within the past 30 days, and having documented chronic diseases or congenital abnormalities or malformation were excluded from this study. Based on urine culture results, the febrile UTI group was further categorized into *Escherichia coli* and non-*E. coli* subgroups. The flow chart of inclusion and exclusion criteria is shown in Figure 1.

Febrile infants with negative results on urine and blood culture samples were recruited as the febrile control group. Virus isolation results revealed etiology as parainfluenza, respiratory syncytial virus, adenovirus, or enterovirus infection in the febrile control group. Empirical antibiotics were given to all febrile infants for the first three days following admission before urine or blood culture identified or excluded bacterial infection. In addition, 10 healthy infants aged 1 month to 4 months old were recruited to evaluate the urinary hepcidin–creatinine ratio to increase the intensity of the results.

### 2.2. Laboratory Tests and Examinations

Patients’ baseline characteristics were recorded, including age at admission, sex, the occurrence of bacteremia, and preadmission fever duration. Blood analysis included complete blood count with white blood count (WBC) differential count (CBC/DC), serum C-reactive protein (CRP), procalcitonin (PCT), iron, ferritin, total iron binding capacity (TIBC), hepcidin, and blood culture. Urinalysis included measurement of the microscopic exam, urine culture, hepcidin, and creatinine. First blood and urine samples were collected for analysis in the emergency department before initiation of antibiotic treatment. Three days after antibiotic treatment, second blood and urine collection for CBC/DC, serum CRP, PCT, urinalysis, urine culture, hepcidin, and creatinine were performed. Serum and urine samples were stored at −80 °C before analysis, and hepcidin 25-isoforms were measured by using a commercial enzyme-linked immunosorbent assay kit (Catalog Number MBS2516119, Detection Range 1.56–100 ng/mL, MyBioSource, Inc., San Diego, CA, USA) according to the manufacturer’s instructions. The intra- and interassay coefficients of variability (CV) for hepcidin were less than 6.3% and 5.3% respectively. Considering the variation in urine dilution following intravenous fluid administration, urinary hepcidin concentrations were normalized to urinary creatinine. Due to parents’ concerns about the invasive nature of suprapubic aspiration and urinary catheterization, all urine specimens were obtained via a sterile catch-bag method. Urinary tract infection was diagnosed when 100,000 or more colony-forming units (CFU)/mL of a single bacterial strain were detected on the urine culture combined with a positive dipstick and microscopic urinalysis [21].

### 2.3. Statistical Analysis

All parameters were compared between febrile UTI and control groups and *E. coli* and non-*E. coli* groups. Categorical and continuous data were analyzed by Chi-square or Fisher’s exact test and Student’s *t*-test, respectively. In addition, the urinary hepcidin–creatinine ratios of 10 healthy, 10 febrile control, and 10 febrile UTI infants were compared by Kruskal–Wallis Test. Blood and urinary biomarkers on admission and 3 days after antibiotic treatment were compared by paired *t*-test in each group. Univariate and multivariate logistic regression, receiver operating characteristic (ROC) curve analysis, and Spearman’s rank correlation coefficient were used for further analysis of the variables. All statistical tests were performed using IBM SPSS 25 statistics software (IBM Corp., Armonk, NY, USA), and *p* < 0.05 was considered statistically significant.

## 3. Results

### 3.1. Characteristics of Study Population

We recruited a total of 118 febrile infants aged one to four months in this study. Eighty-six infants were diagnosed with febrile UTI, and thirty-two febrile infants with negative blood, urine, and stool culture results were included in the febrile control group. Among the febrile UTI patients, there were 69 infants with *E. coli* UTI, 17 with non-*E. coli* UTI, and 6 with bacteremia. The baseline demographic and lab data of the febrile UTI group and control group are shown in Table 1. No significant differences were found in age, sex, fever duration on admission, or incidence of bacteremia between the two groups.

### 3.2. Analysis between the Febrile UTI and Febrile Control Groups

The first analysis focused on laboratory data between the febrile UTI and febrile control groups (Table 2). On admission, the febrile UTI group had higher WBC, absolute neutrophil count (ANC), serum CRP, procalcitonin, urine WBC and red blood cell (RBC), and urinary hepcidin–creatinine ratio than the fever control group (*p* < 0.001, <0.001, <0.001, 0.008, <0.001, 0.001, and 0.016, respectively). However, a significant reduction in serum iron level was seen in the febrile UTI group compared to the febrile control counterpart (25.03 ± 9.72 vs. 48.55 ± 25.41 mcg/dL, *p* < 0.001). After 3 days of antibiotics treatment, WBC, ANC, RBC, hemoglobin (Hb), mean cell volume (MCV), serum CRP, and urinary hepcidin–creatinine ratio were all showing reduction in the febrile UTI group (*p* < 0.001, <0.001, 0.001, <0.001, <0.001, <0.001, and 0.009, respectively). In the fever control group, only ANC had a significant reduction after 3 days of antibiotic treatment. Insignificant differences in serum hepcidin levels before or after antibiotic treatment were observed in the febrile UTI and febrile control groups. We conducted univariate logistic regression analysis, and the urinary hepcidin–creatinine ratio had an odds ratio of 2.01 (95% confidence interval: 1.089–3.720). In multivariate analysis, there was no significant difference between WBC, CRP, and urine hepcidin before antibiotic treatment. ROC curve analysis demonstrated that the area under the curve for serum WBC, CRP, and urine hepcidin was 0.931, 0.894, and 0.894, respectively. Further ROC curve analysis did not show any significant difference in performance between these variables. Spearman’s correlation coefficient analysis showed a moderate correlation between urine hepcidin and serum WBC before antibiotic treatment (*p* = 0.037).

#### Analysis between the Healthy, Febrile Control, and Febrile UTI Groups

The urinary hepcidin–creatinine ratio was significantly different between the healthy, febrile control, and febrile UTI groups (*p* = 0.001). An increasing trend was observed from healthy to febrile UTI groups (Figure 2).

### 3.3. Analysis between the E. coli and Non-E. coli Groups

The relationship between urinary hepcidin–creatinine ratio and the microbial etiology in the febrile UTI group is shown in Table 3. Among febrile UTI patients, 69 yielded *E. coli* on urine culture and 17 non-*E. coli* infection, including *Klebsiella pneumonia*, Group B *Streptococcus*, and *Enterococcus.* On admission, parameters in blood and urine were not significantly different in the two groups, except hematuria was more significant in the *E. coli* group than non-*E. coli* groups. Hemoglobin in both *E. coli* and non-*E. coli* UTI groups significantly decreased after 3 days of antibiotic treatment (*p* < 0.001 and *p* = 0.040, respectively), and patients with *E. coli* UTI had a significant decrease in urinary hepcidin–creatinine ratio (*p* = 0.030) whereas the non-*E. coli* group showed insignificant change. A difference in serum hepcidin was not found between *E. coli* and non-*E. coli* groups.

## 4. Discussion

In this prospective observational study, clinical pediatric data on serum and urinary hepcidin before and after 3 days of antibiotic treatment and their association with anemia-related parameters in febrile UTI were analyzed. Focusing on young infants, patients with febrile UTI had elevated urinary hepcidin–creatinine ratios compared to the non-UTI group on admission which decreased 3 days after antibiotic treatment. Moreover, hypoferremia was noted upon emergency department admission and developed a decreasing trend of Hb level as early as 3 days after antibiotic treatment. Further analysis comparing *E. coli* and non-*E. coli* UTI showed a significant decreasing trend of urinary hepcidin–creatinine ratio 3 days after antibiotic treatment in the *E. coli* UTI group.

The primary aim of our study was to investigate the occurrence of anemia of inflammation and its association with hepcidin in the infant population. We found elevated serum hepcidin level in both the febrile UTI and febrile control group, but evidence of anemia of inflammation were only seen in febrile UTI patients, such as concurrent decreased Hb levels and hypoferremia. This may reflect the host’s iron sequestration strategy in response to the bacterial invasion. Iron is vital for many functional proteins involving oxygen transport and energy production and thus an essential component for survival among humans and nearly all organisms [22]. During acute infection, microbes have evolved strategies to obtain iron from the host. Meanwhile, human innate immunity has evolved defense mechanisms to restrict the accessibility of iron from pathogens such as decreased absorption of dietary iron and release of iron from macrophages, hepatocytes, and enterocytes by upregulating hepcidin, a disulfide-rich antimicrobial peptide [23]. The induction of hepcidin expression during inflammation can result in a rapid reduction in serum iron concentrations, and persistent high levels of hepcidin and hypoferremia may lead to anemia of inflammation [22]. However, we did not observe decreased Hb levels on admission in this study. According to previous research, estimated lifespan of neonatal erythrocytes was around 54.2 days [24]. If erythropoiesis is abated by hypoferremia, it would take approximately five days in theory to reduce Hb levels by 10%, a decrease to be considered clinically relevant and detectable. Because all patients in this study were admitted with fever less than one day, a difference in Hb levels was not detectable on admission or three days afterward. However, a decreasing trend in Hb levels was observed in the febrile UTI group three days after admission, which could potentially lead to anemia if we followed the hemogram further on. These results are consistent with previous studies by Kossiva et al. [3] and van Eijk et al. [25], both of which showed a decrease in Hb levels following hepcidin elevation during infection.

Despite observing concurrent elevated serum hepcidin, decreased serum iron level, and decreasing trend in Hb levels, our Spearman’s correlation coefficient analysis did not identify a significant correlation between serum hepcidin levels and serum iron levels or Hb levels in either the febrile UTI group or febrile control group. These results suggest that while hepcidin may be involved in the regulation of iron metabolism during infection in infancy, its relationship with serum iron and Hb levels may be complex and multifactorial. Indeed, previous studies have reported a hepcidin-independent mechanism of hypoferremia under inflammatory stimulation [26,27]. Both bacterial and viral infection would elevate serum hepcidin levels, while hypoferremia has only been seen in bacterial infection [3,28,29]. Consistent with previous studies, no significant difference in serum hepcidin but significant hypoferremia and associated anemia were found in febrile UTI in this study.

The production of hepcidin in hepatocytes is stimulated by various factors, including high extracellular iron level or infectious or inflammatory stimulus. Elevated serum hepcidin level in combination with Hb-for-age *Z*-score were found to be predictive for Kawasaki disease [30]. In patients with acute leukemia or undergoing hematopoietic cell transplantation who frequently received red blood cell concentrate transfusions, serum ferritin and hepcidin levels were upregulated at the same time by systemic iron overload [31]. Furthermore, higher levels of serum hepcidin level was noted in obese children and adolescents, which may suggest a low-grade inflammatory state and may reflect in increased iron content in these patient groups [32].

Our study demonstrated concurrent elevation of serum and urinary hepcidin on admission, but only urinary hepcidin showed significant decrease three days after antibiotic treatment, while serum hepcidin remained elevated. Similarly, serum WBC and CRP levels also had a significant decrease three days after antibiotic treatment, but only serum WBC before antibiotic treatment showed moderate correlation with urinary hepcidin (*p* = 0.037). No significant correlation was observed between the changes of CRP and changes of urinary hepcidin. Additionally, we explored the relationship of urinary hepcidin with anemia-related parameters. Despite a decreasing trend in Hb levels similar to that seen in urinary hepcidin levels and serum iron levels that decreased significantly on admission, we did not find a statistically significant correlation between urinary hepcidin and either Hb or serum iron. These findings were compatible with previous studies in Crohn’s disease and older persons with proinflammatory states, which reported no correlation between urinary hepcidin and CRP [33,34]. In addition, a study in anemic school children with Schistosoma haematobium demonstrated a significant correlation between urinary hepcidin and CRP but not Hb levels [35].

Hepcidin is a small protein, and after filtration across the glomerulus, it is almost completely reabsorbed in the proximal tubule. Its urinary faction excretion is estimated to be between 0% and 3% [36]. In addition, researchers found renal expression of hepcidin at the apical pole of the thick ascending limb and collecting tubules [37]. This indicates that hepcidin may be synthesized locally in the kidney and then released into the urine. Clinical studies on the relevance of urinary hepcidin are just emerging. In adult cardiac surgery cohorts, urinary hepcidin was evaluated as a diagnostic biomarker for acute kidney injury, and its elevation discriminated patients who did not develop acute kidney injury. In our study, an elevation of urinary hepcidin–creatinine ratio in the febrile UTI group on admission was found. We explored the ability of urine hepcidin in predicting UTI among febrile infants, and ROC curve analysis did not show significant difference in performance between urine hepcidin, serum WBC, and CRP. This suggests that urine hepcidin may not be inferior to the other two makers in predicting UTI.

Possible causes of the elevated urinary hepcidin–creatinine ratio in UTI may be due to decreased tubular reabsorption, or local synthesis, or both. Moreover, hematuria was also more prominent in the febrile UTI group. Red blood cells in the urine release Hb and heme-related products. Once taken up by tubular cells, Hb dissociates and releases the heme group, which induces oxidative stress, cell death, and inflammation. It is unknown whether local renal hepcidin exerts protective effects in face of the oxidative stress of hematuria in febrile UTI patients.

Hepcidin, as an antimicrobial peptide, has broad antimicrobial activity against Gram-positive bacteria (e.g., *Staphylococcus aureus*, *Staphylococcus epidermidis*, *Enterococcus faecium*), Gram-negative bacteria (e.g., *Pseudomonas aeruginosa*, *Escherichia coli*, *Acinetobacter baumannii*), and fungi (e.g., *Candida albicans* and *Saccharomyces cerevisiate*) without cytotoxicity in mammalian cells [38,39,40]. A previous animal study revealed that hepcidin could be an effective component of renal defense systems against uropathogenic *E. coli* infection via the mechanisms of urinary iron restriction, urine acidification, renal inflammatory response, and its antimicrobial activity [16]. Additionally, uropathogenic *E. coli* could repress renal hepcidin production 24 h post-*E. coli* infection as a strategy to evade the antibacterial activity of hepcidin [16], which could have clinical relevance. In our results, the urinary hepcidin–creatinine ratio decreased more in the *E. coli* UTI group 3 days after antibiotic treatment than that of the non-*E. coli* UTI group. We postulated that this was likely caused by *E. coli* locally repressing renal hepcidin synthesis as an escape strategy against human innate immunity. In contrast, urine cultures 3 days after antibiotics treatment all yielded negative results. This may indicate that decreased *E. coli* bacterial load leads to decreased urinary hepcidin–creatinine ratios. However, further animal studies will be required to clarify the relationship between bacterial load and urinary hepcidin level.

To our knowledge, no prior studies have examined the phenomenon of anemia of inflammation and its association with hepcidin levels in the infant population. Serum and urinary hepcidin levels and Hb were investigated simultaneously to detect whether there is an interrelationship between them. Pediatric studies defining reference ranges for serum and urinary hepcidin are limited. Previous studies showed age differences in hepcidin levels, and Aranda et al. found that the hepcidin level increased in healthy infants during the first year of life [18]. In our prospective study with serial samples, we used a febrile age-matched control group to minimize the potential impact of age on hepcidin levels. We recruited a healthy control group to evaluate the impact of infection on hepcidin levels as well. Nevertheless, the study has several limitations. One limitation is that the urine samples were collected via perineal bags, not via urine catheter or suprapubic puncture, and may be more easily contaminated. However, only the urine culture results with 100,000 or more CFU/mL of a single bacterial strain combined with pyuria in urinalysis were considered UTI in this study. Furthermore, compared with the procedure of perineal bags, suprapubic puncture is more invasive and, though rare, may lead to complications, e.g., bowel or large vessel puncture, leakage of urine into the tissue, and abscess formation, all which are great concerns to the parents and health care providers [41]. Moreover, invasive procedures may impose stress on children, with possible negative psychological consequences [42]. Thus, it is our policy to use perineal bag instead of suprapubic puncture to collect the urine. Furthermore, our study includes only febrile infants from a single tertiary hospital, and future studies conducted in a multicenter setting may be warranted.

## 5. Conclusions

The urinary hepcidin–creatinine ratio had the highest augmentation during the acute infection in febrile UTI patients, especially in *E. coli* UTI. Significant hypoferremia and a decreasing trend of Hb were also found in these patients. Thus, urinary hepcidin might play a pathological role in anemia of febrile UTI infants. We identified the occurrence of anemia and an additive value of urinary hepcidin in infants with UTI. Future research to investigate the diagnostic and prognostic values of urinary hepcidin will be necessary.

## Figures and Tables

**Figure 1 children-10-00870-f001:**
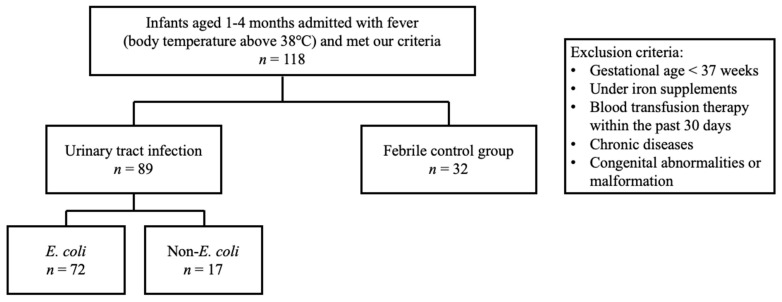
The flowchart of study selection.

**Figure 2 children-10-00870-f002:**
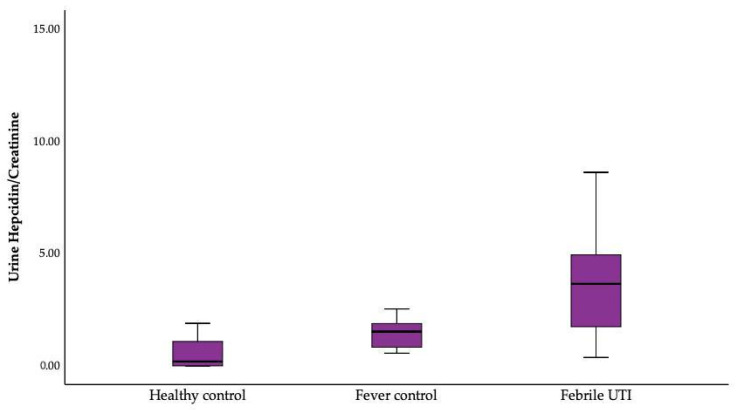
Urinary hepcidin–creatinine ratio analyzed between the healthy, febrile control, and febrile UTI groups by Kruskal–Wallis Test.

**Table 1 children-10-00870-t001:** The characteristics of patients with febrile urinary tract infection and fever control.

	Febrile UTI (*n* = 86)	Fever Control (*n* = 32)	*p* Value
Age (day)	75.03 ± 26.07	76.31 ± 20.43	0.779
Sex (F/M)	24/65	13/19	0.113
Bacteremia (*n*, %)	6, 7%	0, 0%	0.151
Fever duration at admission (day)	0.84 ± 0.69	0.78 ± 0.52	0.676

Continuous variables were analyzed by Student’s *t*-test of variance and presented as mean ± standard deviation. Categorical variables were analyzed by the Chi-square test or Fisher’s exact test. F/M: female/male.

**Table 2 children-10-00870-t002:** The blood and urinary laboratory data before and 3 days after antibiotic treatment.

	Febrile UTI (*n* = 86)		Fever Control (*n* = 32)		*p* Value	Logistic Regression	
						OR (95% C.I.)	*p* Value
Blood Laboratory		* *p* Value		* *p* Value			
WBC (1000/μL)		<0.001		0.24			
Before	14.61 ± 4.80		8.21 ± 3.05		<0.001	1.46 (1.26–1.70)	<0.001
After	10.27 ± 3.66		8.96 ± 2.79		0.103	1.14 (0.98–1.33)	0.093
ANC (1000/μL)		<0.001		<0.001			
Before	51.97 ± 13.19		40.64 ± 16.50		<0.001	1.06 (1.03–1.09)	<0.001
After	23.64 ± 8.94		21.65 ± 12.18		0.373	1.02 (0.97–1.07)	0.376
RBC (million/μL)		0.001		0.176			
Before	3.72 ± 0.57		3.70 ± 0.43		0.833	1.02 (0.47–2.22)	0.955
After	3.64 ± 0.48		3.63 ± 0.46		0.917	0.99 (0.37–2.62)	0.986
Hb (g/dL)		<0.001		0.083			
Before	10.48 ± 1.23		10.53 ± 0.86		0.82	0.94 (0.66–1.33)	0.717
After	10.10 ± 0.94		10.18 ± 0.91		0.704	0.88 (0.54–1.44)	0.619
MCV (fL)		<0.001		0.13			
Before	85.66 ± 7.30		85.96 ± 5.79		0.918	0.99 (0.94–1.06)	0.968
After	83.16 ± 6.57		84.80 ± 4.90		0.26	0.96 (0.89–1.04)	0.282
CRP (mg/L)		<0.001		0.12			
Before	42.55 ± 40.80		5.36 ± 6.45		<0.001	1.16 (1.08–1.25)	<0.001
After	12.63 ± 15.52		4.03 ± 4.28		<0.001	1.17 (1.04–1.31)	0.007
Procalcitonin (ng/mL)		0.998		0.51			
Before	1.88 ± 4.81		0.13 ± 0.09		0.008	3522.36 (10.73–1,156,353.46)	0.006
After	2.83 ± 10.91		0.12 ± 0.09		0.23	53.65 (0.95–3028.83)	0.053
Fe (ng/mL)	25.03 ± 9.72		48.55 ± 25.41		<0.001	0.88 (0.82–0.94)	<0.001
Ferritin (ng/mL)	165.38 ± 99.98		233.70 ± 187.56		0.094	0.99 (0.99–1.00)	0.114
TIBC (ug/dL)	294.58 ± 45.85		295.10 ± 50.67		0.968	1.00 (0.99–1.01)	0.967
Hepcidin (ng/mL)		0.907		0.744			
Before	239.01 ± 41.23		229.0 ± 35.66		0.51	1.00 (0.99–1.03)	0.495
After	242.65 ± 52.94		239.49 ± 60.37		0.893	1.00 (0.99–1.02)	1.887
**Urinary laboratory**							
WBC (/μL)	314.57 ± 145.09		4.69 ± 6.81		<0.001	1.16 (1.063–1.261)	0.001
RBC (/μL)	87.12 ± 178.57		1.22 ± 2.73		0.001	1.39 (1.174–1.652)	<0.001
Creatinine (mg/dL)		0.257		0.698			
Before	11.06 ± 8.18		12.92 ± 5.88		0.577	0.97 (0.88–1.08)	0.566
After	11.61 ± 7.02		9.31 ± 6.21		0.436	1.06 (0.91–1.24)	0.43
Hepcidin/Creatinine		0.009		0.259			
Before	11.06 ± 6.93		4.22 ± 2.26		0.016	2.01 (1.089–3.720)	0.026
After	7.49 ± 4.41		3.73 ± 2.77		0.041	1.37 (0.997–1.886)	0.052

All variables were analyzed by Student’s *t*-test between febrile urinary tract infection (UTI) and fever control groups, expressed as *p* value. Data before and after 3 days of antibiotic treatment were analyzed by paired *t*-test and expressed as * *p* value. The relationship of each variable with febrile UTI was analyzed by univariate logistic regression, expressed as odds ratio (OR) with 95% confidence interval (CI) and *p* value. All data are presented as mean ± standard deviation. ANC: absolute neutrophil count, C.I.: confidence interval, CRP: C-reactive protein, Hb: hemoglobin, MCV: mean cell volume, OR: odds ratio, RBC: red blood cell, TIBC: total iron binding capacity, UTI: urinary tract infection, WBC: white blood cell.

**Table 3 children-10-00870-t003:** The laboratory data before and after 3 days of antibiotic treatment in *E. coli* and non-*E. coli* urinary tract infection.

	Urinary Tract Infection				*p* Value	Logistic Regression	
	*E. coli* (*n* = 69)		Non-*E. coli* (*n* = 17)			OR (95% C.I.)	*p* Value
**Blood laboratory**		*** *p* value**		*** *p* value**			
WBC (1000/μL)		<0.001		0.007			
Before	14.96 ± 4.50		13.09 ± 5.83		0.23	1.09 (0.97–1.22)	0.147
After	10.22 ± 3.61		9.66 ± 3.15		0.464	1.07 (0.91–1.26)	0.413
ANC (1000/μL)		<0.001		0.001			
Before	52.11 ± 13.69		51.42 ± 11.14		0.846	1.00 (0.96–1.05)	0.831
After	24.27 ± 8.75		21.79 ± 9.69		0.116	1.06 (0.99–1.13)	0.121
RBC (million/μL)		0.005		0.083			
Before	3.76 ± 0.55		3.55 ± 0.65		0.175	1.98 (0.67–5.80)	0.214
After	3.69 ± 0.45		3.43 ± 0.55		0.049	3.63 (0.919–14.347)	0.066
Hb (g/dL)		<0.001		0.04			
Before	10.54 ± 1.20		10.22 ± 1.47		0.339	1.23 (0.77–1.97)	0.388
After	10.18 ± 0.96		9.73 ± 0.77		0.079	1.78 (0.90–3.53)	0.1
MCV (fL)		<0.001		<0.001			
Before	85.37 ± 7.05		87.72 ± 7.79		0.229	0.96 (0.88–1.03)	0.253
After	82.59 ± 6.28		85.58 ± 7.39		0.101	0.93 (0.84–1.02)	0.114
CRP (mg/L)		<0.001		0.001			
Before	42.14 ± 39.79		44.25 ± 46.09		0.85	1.00 (0.99–1.01)	0.797
After	13.31 ± 15.99		9.72 ± 13.41		0.408	1.02 (0.97–1.07)	0.418
Procalcitonin (ng/mL)		0.996		0.961			
Before	2.11 ± 5.36		1.03 ± 1.31		0.496	1.08 (0.87–1.34)	0.5
After	3.02 ± 11.70		1.61 ± 2.89		0.815	1.02 (0.88–1.18)	0.804
Fe (ng/mL)	23.41 ±7.86		34.00 ± 15.12		0.257	0.89 (0.79–1.00)	0.068
Ferritin (ng/mL)	142.86 ± 67.96		289.23 ± 163.70		0.171	0.99 (0.97–0.99)	0.032
TIBC (ug/dL)	287.18 ± 36.23		335.25 ± 75.66		0.295	0.98 (0.95–1.00)	0.078
Hepcidin (ng/mL)		0.839		0.616			
Before	242.26 ± 43.34		219.5 ± 23.33		0.492	1.02 (0.97–1.06)	0.463
After	248.14 ± 54.80		212.40 ± 37.90		0.403	1.02 (0.98–1.06)	0.377
**Urinary laboratory**							
WBC (/μL)	333.21 ± 180.24		235.65 ± 208.50		0.055	1.00 (1.00–1.00)	0.065
RBC (/μL)	103.11 ± 156.74		19.41 ± 27.17		<0.001	1.02 (0.997–1.036)	0.09
Creatinine (mg/dL)		0.838		0.493			
Before	11.15 ± 8.69		10.52 ±5.10		0.89	1.01 (0.88–1.16)	0.884
After	10.90 ± 6.12		15.68 ±11.25		0.215	0.92 (0.80–1.05)	0.224
Hepcidin/Creatinine		0.03		0.102			
Before	11.17 ± 7.20		10.42 ± 5.89		0.847	1.02 (0.86–1.20)	0.84
After	7.84 ± 4.66		5.46 ± 1.72		0.329	1.17 (0.86–1.61)	0.322

All variables were analyzed by Student’s *t*-test between *E. coli* urinary tract infection (UTI) and non-*E. coli* UTI groups, expressed as *p* value. Data before and after 3 days of antibiotic treatment were analyzed by paired *t*-test and expressed as * *p* value. The relationship of each variable with *E. coli* UTI was analyzed by univariate logistic regression, expressed as odds ratio (OR) with 95% confidence interval (CI) and *p* value. All data are presented as mean ± standard deviation. ANC: absolute neutrophil count, C.I.: confidence interval, CRP: C-reactive protein, Hb: hemoglobin, MCV: mean cell volume, OR: odds ratio. RBC: red blood cell, TIBC: total iron binding capacity, UTI: urinary tract infection, WBC: white blood cell.

## Data Availability

Data are unavailable due to privacy or ethical restrictions.

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
