# Peer review of "Elevated Urinary Hepcidin Level and Hypoferremia in Infants with Febrile Urinary Tract Infection: A Prospective Cohort Study"

_children, 2023, doi:10.3390/children10050870_

Round 1

Reviewer 1 Report

The authors reported on Hepcidin, a potential new marker for febrile urinary tract infections in infants. Urinary Hepcidin can be a simple useful marker for infants with f-UTI.

I have some questions for the authors.

1. The authors diagnosed febrile urinary tract infection based on blood or urine culture results. Did the authors confirm that the patients diagnosed with f-UTI had a true urinary tract infection by contrast-enhanced CT or renal ultrasound?  If supported by definitive imaging studies, hepcidin would be a very useful diagnostic marker for f-UTI. Is it possible that patients with positive E. coli had a true UTI?

2. The authors reported that hepcidin was significantly elevated in the f-UTI group. Did the authors perform a multivariate analysis? You need to show whether Hepcidin was a more significant marker than WBC and CRP.

3. Did the authors assess renal function with DMSA scan? If there is an association between hepcidin and abnormal DMSA findings, it could be an excellent marker for diagnosing clinically significant UTIs.

Author Response

Point 1: The authors diagnosed febrile urinary tract infection based on blood or urine culture results. Did the authors confirm that the patients diagnosed with f-UTI had a true urinary tract infection by contrast-enhanced CT or renal ultrasound? If supported by definitive imaging studies, hepcidin would be a very useful diagnostic marker for f-UTI. Is it possible that patients with positive E. coli had a true UTI?
Response 1: Thank you for your comment.
All the UTI patients were diagnosed with significant bacteriuria, defined as urine culture with 100,000 or more colony-forming unit/mL of a single uropathogen, combined with an inflammatory response, indicated by the presence of pyuria on dipstick and microscopic urinalysis in accordance with international guidelines as those from the American Academy of Pediatrics (AAP) [1] or the National Institute for Health and Care Excellence (NICE) group [2]. (The above illustrations have been
incorporated in Line 116-119)

We conducted kidney ultrasound for every enrolled infant diagnosed with UTI to assess the extent of the disease, whether it involved only the lower urinary tract or with renal involvement. Among the 89 UTI patients, 53 of them (59.6%) had kidney ultrasound results compatible with acute pyelonephritis. Following the NICE [2] and AAP [1] guidelines, we reserved DMSA scan for patients with recurrent or atypical UTI to evaluate the presence of renal scarring at follow-up. In order to minimize radiation
exposure and since CT exams are not recommended for initial evaluation of children with first UTI, we did not routinely perform contrast-enhanced CT exams.

Comment 2: The authors reported that hepcidin was significantly elevated in the f-UTI group. Did the authors perform a multivariate analysis? You need to show whether Hepcidin was a more significant marker than WBC and CRP.
Response 2: Thank you for your comment.
Indeed, we conducted multivariate analysis. However, there was no significant difference between WBC, CRP and urine hepcidin before antibiotic treatment. ROC curve analysis demonstrated that the area under the curve for serum WBC, CRP and urine hepcidin was 0.931, 0.894, and 0.894, respectively. Further ROC curve analysis did not show any significant difference in performance between these variables, suggesting that urine hepcidin may not be inferior to serum WBC or CRP in predicting UTI among febrile infants. (The above illustrations have been incorporated in Line 288-291)

Point 3: Did the authors assess renal function with DMSA scan? If there is an association between hepcidin and abnormal DMSA findings, it could be an excellent marker for diagnosing clinically significant UTIs.

Response 3: Thank you for your comment.
We performed kidney ultrasound for every enrolled infant diagnosed with UTI and reserved Dimercaptosuccinic acid (DMSA) kidney scan for children with recurrent or atypical UTI, in accordance with the guidelines from the AAP [1] and NICE group [2]. We also considered DMSA exam if the ultrasound findings indicated the presence of vesiculoureteral refulx (VUR). In our study, only 2 out of the 89 UTI patients received DMSA scan because of recurrent UTI. However, in our future study, we would consider including DMSA in our study protocol to evaluate the association between renal function and hepcidin.
References
1. Roberts, K.B. Urinary tract infection: clinical practice guideline for the diagnosis and management of the initial UTI in febrile infants and children 2 to 24 months. Pediatrics 2011, 128, 595-610, doi:10.1542/peds.2011-1330.

2. National Collaborating Centre for, W.s.; Children's, H. National Institute for Health and Clinical Excellence: Guidance. In Urinary Tract Infection in Children: Diagnosis, Treatment and Long-term Management; RCOG Press Copyright © 2007, National Collaborating Centre for Women’s and Children's Health.: London, 2007.

Reviewer 2 Report

The authors presented a well conducted and well written study regarding the presence of elevated urinary hepcidin level and hypoferremia in infants with febrile urinary tract infection. I have only some comments which may improve the quality of the study.

1. It is important to show whether there was a link of elevated urinary hepcidin level with hypoferremia and anemia. Was there any correlation of urinary and/or serum hepcidin with iron profile and Hb in patients with febrile UTI and in febrile patients without UTI? Please add in the manuscript.

 2. Moreover, it would be interesting to investigate the correlation between CRP and urinary hepcidin as well as the correlation between changes of urinary hepcidin and  changes of CRP before and after treatment in febrile UTI patients.

3. Why did the authors choose only infants aged 1-4 months as patient population? Please explain in the manuscript.

Author Response

Point 1: It is important to show whether there was a link of elevated urinary hepcidin level with hypoferremia and anemia. Was there any correlation of urinary and/or serum hepcidin with iron profile and Hb in patients with febrile UTI and in febrile patients without UTI? Please add in the manuscript.
Response 1: Thank you for your comment.
Our findings revealed that both febrile UTI and febrile control groups exhibited an increase in serum hepcidin levels, but only the febrile UTI group demonstrated a statically significant decrease in serum iron level, and a decreasing trend in Hb levels. However, despite observing these changes, our Spearman’s correlation coefficient analysis did not identify a significant correlation between serum hepcidin levels and serum iron levels or Hb levels in either the febrile UTI group or febrile control
group. These results suggest that while hepcidin may be involved in the regulation of iron metabolism during infection in infancy, its relationship with serum iron and Hb levels may be complex and multifactorial. Indeed, previous studies have reported a hepcidin-independent mechanism of
hypoferremia under inflammatory stimulation [1,2]. Both bacterial and viral infection would elevate serum hepcidin levels while hypoferremia has only seen in bacterial infection [3-5]. Consistent with previous studies, no significant difference in serum hepcidin but significant hypoferremia and
associated anemia were found in febrile UTI in this study. Additionally, we explored the relationship of urinary hepcidin with anemia-related parameters. Despite a decreasing trend in Hb levels similar to that seen in urinary hepcidin levels, and serum iron levels decreased significantly on admission, we did not find a statistically significant correlation between urinary hepcidin and either Hb or serum iron. However, a study in anemic school children with Schistosoma haematobium demonstrated a significant correlation between urinary hepcidin and CRP, but not Hb levels [6]. (The above illustrations have been incorporated in line 218-254, 270 278)

Point 2: Moreover, it would be interesting to investigate the correlation between CRP and urinary hepcidin as well as the correlation between changes of urinary hepcidin and changes of CRP before and afte treatment in febrile UTI patients.
Response 2: Thank you for your comment.
In Spearman’s correlation coefficient analysis, we did not find correlation between CRP and urinary hepcidin. Also, no correlation between changes of urinary hepcidin and changes of CRP before and after treatment were noted. However, we observed moderate correlation between urinary hepcidin and WBC before antibiotic treatment (p = 0.037). The findings are compatible with previous studies in Crohn’s disease and older persons with proinflammtory state that urinary hepcidin was not correlated with markers of inflammation [7,8]. (The above illustrations have been incorporated in Line 264-270)

Point 3: Why did the authors choose only infants aged 1-4 months as patient population? Please explain in the manuscript.
Response 3: Thank you for your comment.
We specifically selected infants in this age group to ensure a relatively homogenous population, taking into consideration the known age-related difference in hepcidin [9,10]. In addition, we excluded neonates with UTI as they are often associated with congenital anomalies of the kidney and urinary tract (CAKUT) [11,12]. Furthermore, neonates were not included as changes in Hb levels in this group may be influenced by factors such as polycythemia, hemodilution with somatic growth, alloimmune hemolytic disease of the newborn, or nutritional anemia. (The above illustrations have been incorporated in Line 77-83)

References
1. Guida, C.; Altamura, S.; Klein, F.A.; Galy, B.; Boutros, M.; Ulmer, A.J.; Hentze, M.W.; Muckenthaler, M.U. A novel inflammatory pathway mediating rapid hepcidin-independent hypoferremia. Blood 2015, 125, 2265-2275, doi:10.1182/blood-2014-08-595256.
2. Kim, A.; Fung, E.; Parikh, S.G.; Valore, E.V.; Gabayan, V.; Nemeth, E.; Ganz, T. A mouse model of anemia of inflammation: complex pathogenesis with partial dependence on hepcidin. Blood 2014, 123, 1129-1136, doi:10.1182/blood-2013-08-521419.

3. Oppen, K.; Ueland, T.; Siljan, W.W.; Skadberg, Ø.; Brede, C.; Lauritzen, T.; Aukrust, P.; Steinsvik, T.; Husebye, E.; Michelsen, A.E.; et al. Hepcidin and Ferritin Predict Microbial Etiology in Community-Acquired Pneumonia. Open Forum Infect Dis 2021, 8, ofab082, doi:10.1093/ofid/ofab082.
4. Kossiva, L.; Gourgiotis, D.I.; Tsentidis, C.; Anastasiou, T.; Marmarinos, A.; Vasilenko, H.; Sdogou, T.; Georgouli, H. Serum hepcidin and ferritin to iron ratio in evaluation of bacterial versus viral infections in children: a single-center study. Pediatr Infect Dis J 2012, 31, 795-798, doi:10.1097/INF.0b013e318256f843.
5. Kossiva, L.; Soldatou, A.; Gourgiotis, D.I.; Stamati, L.; Tsentidis, C. Serum hepcidin: indication of its role as an "acute phase" marker in febrile children. Ital J Pediatr 2013, 39, 25, doi:10.1186/1824-7288-39-25.
6. Ayoya, M.A.; Spiekermann-Brouwer, G.M.; Stoltzfus, R.J.; Nemeth, E.; Habicht, J.P.; Ganz, T.; Rawat, R.; Traoré, A.K.; Garza, C. Alpha 1-acid glycoprotein, hepcidin, C-reactive protein, and serum ferritin are correlated in anemic schoolchildren with Schistosoma haematobium. Am J Clin Nutr 2010, 91, 1784-1790, doi:10.3945/ajcn.2010.29353.
7. Ferrucci, L.; Semba, R.D.; Guralnik, J.M.; Ershler, W.B.; Bandinelli, S.; Patel, K.V.; Sun, K.; Woodman, R.C.; Andrews, N.C.; Cotter, R.J.; et al. Proinflammatory state, hepcidin, and anemia in older persons. Blood 2010, 115, 3810-3816, doi:10.1182/blood-2009-02-201087.
8. Semrin, G.; Fishman, D.S.; Bousvaros, A.; Zholudev, A.; Saunders, A.C.; Correia, C.E.; Nemeth, E.; Grand, R.J.; Weinstein, D.A. Impaired intestinal iron absorption in Crohn's disease correlates with disease activity and markers of inflammation. Inflamm Bowel Dis 2006, 12, 1101-1106, doi:10.1097/01.mib.0000235097.86360.04.
9. Sdogou, T.; Tsentidis, C.; Gourgiotis, D.; Marmarinos, A.; Gkourogianni, A.; Papassotiriou, I.; Anastasiou, T.; Kossiva, L. Immunoassay-based serum hepcidin reference range measurements in healthy children: differences among age groups. J Clin Lab Anal 2015, 29, 10-14, doi:10.1002/jcla.21719.
10. Aranda, N.; Bedmar, C.; Arija, V.; Jardí, C.; Jimenez-Feijoo, R.; Ferré, N.; Tous, M. Serum hepcidin levels, iron status, and HFE gene alterations during the first year of life in healthy Spanish infants. Ann Hematol 2018, 97, 1071-1080, doi:10.1007/s00277-018-3256-2.
11. Bonadio, W.; Maida, G. Urinary tract infection in outpatient febrile infants younger than 30 days of age: a 10-year evaluation. Pediatr Infect Dis J 2014, 33, 342-344, doi:10.1097/inf.0000000000000110.
12. Goldman, M.; Lahat, E.; Strauss, S.; Reisler, G.; Livne, A.; Gordin, L.; Aladjem, M. Imaging after urinary tract infection in male neonates. Pediatrics 2000, 105, 1232-1235, doi:10.1542/peds.105.6.1232
